# The Management of Metastatic Spinal Cord Compression in Routine Clinical Practice

**DOI:** 10.3390/cancers15102821

**Published:** 2023-05-18

**Authors:** Luis Alberto Pérez-Romasanta, Estanislao Arana, Francisco M. Kovacs, Ana Royuela

**Affiliations:** 1Department of Radiation Oncology, Hospital Universitario de Salamanca, Instituto de Investigaciones Biomédicas de Salamanca (IBSAL), 37007 Salamanca, Spain; 2Spanish Back Pain Research Network (REIDE), 28008 Madrid, Spain; estanis.arana@ext.uv.es (E.A.); fmkovacs@kovacs.org (F.M.K.); aroyuela@idiphim.org (A.R.); 3Department of Radiology, Fundación Instituto Valenciano de Oncología, 46009 Valencia, Spain; 4Back Pain Unit, HLA-Moncloa University Hospital, 28008 Madrid, Spain; 5Clinical Biostatistics Unit, Instituto de Investigación Sanitaria Puerta de Hierro-Segovia de Arana, Consorcio de Investigación Biomédica en Red: Epidemiología y Salud Pública (CIBERESP), 28222 Madrid, Spain

**Keywords:** cancer, spinal metastases, metastatic spinal cord compression

## Abstract

**Simple Summary:**

Most Spanish specialists involved in the clinical management of spinal cord compression are familiar with the scoring systems for spine instability and spinal compression as well as with the NICE guideline recommendations. However, many do not apply them in routine practice. Scores on the scales used to evaluate spine instability in neoplastic diseases were interpreted correctly by 57.5–70.0% of the practitioners while scores of the spinal cord compression grading system were interpreted correctly by 30.0–37.5%. There is room for improvement in the management of SMD in routine practice.

**Abstract:**

(1) Background: Whether clinical management of spinal metastatic disease (SMD) matches evidence-based recommendations is largely unknown. (2) Patients and Methods: A questionnaire was distributed through Spanish Medical Societies, exploring routine practice, interpretation of the SINS and ESCC scores and agreement with items in the Tokuhashi and SINS scales, and NICE guideline recommendations. Questionnaires were completed voluntarily and anonymously, without compensation. (3) Results: Eighty specialists participated in the study. A protocol for patients with SMD existed in 33.7% of the hospitals, a specific multidisciplinary board in 33.7%, 40% of radiological reports included the ESCC score, and a prognostic scoring method was used in 73.7%. While 77.5% of the participants were familiar with SINS, only 60% used it. The different SINS and ESCC scores were interpreted correctly by 57.5–70.0% and 30.0–37.5% of the participants, respectively. Over 70% agreed with the items included in the SINS and Tokuhashi scores and with the recommendations from the NICE guideline. Differences were found across private/public sectors, hospital complexity, number of years of experience, number of patients with SMD seen annually and especially across specialties. (4) Conclusions: Most specialists know and agree with features defining the gold standard treatment for patients with SCC, but many do not apply them.

## 1. Introduction

Spinal metastases are the most common type of bone metastasis [1,2]. Spinal metastatic disease can lead to bone fracture, instability, and metastatic spinal cord compression (MSCC). The latter is a devastating complication, which appears in 2.5–10.0% of patients with cancer and 40% of those with bone metastases [3,4]. The prognosis of MSCC is better if it is treated before paresis appears, but 50% of patients lose the ability to walk before they get diagnosed [1,5,6]. 

Timely and efficient coordination among different specialists is paramount for appropriate treatment [2]. To this end, several standardized methods have been developed [7,8,9,10,11]. However, whether these methods are actually used in routine clinical practice is largely unknown. In fact, audits have reported inconsistencies between recommendations of clinical guidelines issued by the National Institute for Health and Care Excellence (NICE) and actual routine practice in the United Kingdom and Ireland [12,13,14]. 

The objective of this study was to explore the management of MSCC in routine practice in Spain and to assess whether it followed available evidence-based recommendations.

## 2. Materials and Methods

A questionnaire was distributed to specialists involved in the clinical management of MSCC.

### 2.1. Subjects

All physicians treating patients with MSCC in Spain were welcome to complete the questionnaire. An invitation was sent to all members of the Spanish scientific societies representing Medical Oncology (“SEOM”—3035 members), Radiation Oncology (“SEOR”—1201 members), Radiology (“SERAM”—6024 members), Neurosurgery (“SENEC”—795 members) Orthopedic Surgery (“SECOT”—5120 members), and clinicians specialized in spine conditions (including neurosurgeons and orthopedic surgeons) (“GEER”—303 members). 

### 2.2. Questionnaire

The questionnaire (Appendix A) gathered information on the participant’s characteristics, work setting, clinical practice, and familiarity with the methods for management of MSCC.

Participants’ characteristics included age (date of birth); medical specialty; seniority (in-training/certified specialist); number of years of clinical practice since certification; and number of patients with MSCC managed during the last 12 months.

Data on work setting included: private/public sector (“National Health Service” or “NHS” if healthcare was funded by taxpayers or “private” if funded by patients or private insurance companies) and data on the hospital in which the clinician worked; ownership (NHS/other governmental institutions/non-profit private institutions/for-profit private companies); management (NHS/private); whether it treated patients covered by the NHS; whether radiological reports quantified compression according to the Epidural Spinal Cord Compression (ESCC) scale [15] (always/occasionally/no); whether a protocol for management of patients with MSCC was implemented and, if so, whether it was multidisciplinary; whether a Board to coordinate care for patients with MSCC existed and, if so, which specialties were included; and hospital complexity (based on number of beds and physicians, academic activity, use of high technology, and performance of highly complex procedures, according to the classification established by the Spanish National Health Service, where Category 1 is the simplest and Category 5 is the most complex) [16].

Data on clinical practice related to MSCC included: method/s used to predict life-expectancy of patients with MSCC, if any (Tokuhashi—original or modified/Bauer—original or modified/Tomita/van der Linden/other); imaging procedure/s used to assess MSCC (entire spine MRI/MRI of the vertebral segment involved/scanner/scanner of the segment involved/other); familiarity with the Spine Instability Neoplastic Score (SINS) [17]; and use of SINS in routine practice (systematically/occasionally/no).

Participants were also asked to interpret the meaning of “1b” and “2” scores on the ESCC scale and “3”, “10”, and “15” on the SINS. 

Finally, respondents were requested to rate their degree of agreement (from 1—strongly disagree to 5—strongly agree) with 18 statements; six focused on the prognostic value of items included in the modified Tokuhashi scale [18] (oncological prognosis, number of spinal metastases, score in general performance tools such as Karnofsky Performance Score or the Eastern Cooperative Oncology Group Score [18,19], visceral metastases, type–location of primary tumor, and degree of paresis). Five statements focused on spine instability [4] (mechanical pain, type of bone lesion—blastic, lytic, or mixed, spinal alignment, degree of vertebral body collapse, and involvement of facet joints). Last seven statements were recommendations from the NICE DG75 clinical guideline for assessment of MSCC (“NICE guideline”) [6,12] (use of MRI, use of full spine MRI to assess MSCC, neurologic examination, assessment by all treating clinicians, and clinical assessment of pain, sphincter control, and limb strength and sensitivity).

### 2.3. Procedure

The authors shared the protocol of the study, but not the questionnaire, with representatives of the SEOM, SEOR, SERAM, SENEC, SECOT, and GEER. These societies forwarded the invitation to participate and a link to the questionnaire to all their members. SENEC and GEER sent an email to all their members, followed by a reminder 1 month later. The other societies published the information in their websites. Members affiliated to two societies (e.g., a neurosurgeon affiliated to SENEC and GEER) were invited twice. 

Participants agreed to complete the questionnaire only once and alone, with no help from other colleagues, and to answer the questions without checking with the literature or colleagues. 

The questionnaire was hosted in Google Forms. No data allowing to identify participants were requested. However, name was requested for those wishing to be informed of the study results. It had been planned that if two respondents coincided in their date of birth and specialty or shared the same IP address, only the first answers introduced would be analyzed. However, this situation did not occur.

Neither respondents nor the scientific societies received any compensation for their contribution to the study.

Results from the questionnaire were stored in an ad hoc database using Microsoft Excel v365.

### 2.4. Analysis

Categorical variables were described by their absolute and relative frequencies. Continuous variables were described by their median, P25, P75, and range values.

Answers on agreement were collapsed into “disagree” (answers 1 to 3) and “agree” (answers 4–5). Answers on clinical practice, interpretation of the SINS and ESCC scores, and agreement with statements were compared across specialties, number of patients treated annually (categorized as ≤7, 8–13, and ≥14), years of experience (categorized as ≤7, 8–13 y, ≥14), private/public sector (working for the NHS vs. privately vs. both), and hospital complexity level (categorized as “simple”—categories 1–3 vs. “complex”—categories 4–5). Comparisons across specialties were restricted to those with ≥5 participants. 

For comparisons, the chi-square or Fisher’s exact tests were used for categorical variables and Mann-Whitney’s U or Kruskall–Wallis tests for numerical ones. Signification was set at 0.05. The statistical package Stata/IC v.16 (StataCorp. 2019. Stata Statistical Software: Release 16. College Station, TX, USA: StataCorp LLC) was used.

## 3. Results

Between 1st June and 30th October 2021, 80 clinicians completed the questionnaire.

### 3.1. Participant’s Characteristics

The typical participant was a 46-year-old certified specialist, who treated annually ≥14 patients with MSCC and had been working for the NHS for 13 years in a grade 4 complexity level hospital, which was owned and managed by the NHS (Table 1).

Management protocols for MSCC were implemented in hospitals where 31 participants (38.8%) worked; thirty were multi-disciplinary. A multidisciplinary board for MSCC existed in 27 (33.8%) hospitals and included between two and six specialties (Table 1). Those more commonly represented were medical oncology, radiation oncology, and orthopedic surgery.

Boards were more common in hospitals where specialists worked both privately and for the NHS (56.0%) than in those where they worked only for the NHS (25.0%) or privately (14.2%) (*p* = 0.020). No other differences related to protocols or boards were found across specialties, number of patients treated annually, years of experience, private/public sector, and hospital complexity level.

### 3.2. Clinical Practice

Full-spine MRI (71.3%) and MRI of the involved segment (22.5%) were the most commonly used imaging procedures for assessing MSCC. Most participants (73.8%) used a prognostic method, although 15.0% used it only occasionally. The Tokuhashi was the most common one (27.5%), but in 33.8% of the hospitals, the scoring system varied across Departments (Table 2). 

The SINS was known by 77.5% of the participants. It was known to more specialists working in “complex” hospitals (88.7%) than in “simple” ones (55.6%) (*p* = 0.002) and to orthopedic surgeons (87.5%) and radiation oncologists (85.0%) than neurosurgeons (66.7%) and medical oncologists (50.0%) (*p* = 0.005).

The SINS was used routinely by 60.0% of specialists. Its use was more common among specialists working in “complex” (71.7%) than “simple” hospitals (37.0%) (*p* = 0.007), among physicians working privately (71.4%) or privately and for the NHS (76.0) than among those working only for the NHS (50.0%) (*p* = 0.019), and among orthopedic surgeons (81.3%), neurosurgeons (66.7%), and radiation oncologists (55.0%) than among medical oncologists (22.2%) (*p* = 0.000).

No other differences in these variables were found across specialties, number of patients treated annually, years of experience, private/public sector, and hospital complexity level.

### 3.3. Accurate Interpretation of Scores

The different SINS and ESCC scores were correctly interpreted by 57.5–70.0% and 30.0–37.5% of the participants, respectively (Table 3). 

The proportion of specialists who interpreted the SINS score correctly was lower among medical oncologists (27.8–44.4%) than among radiation oncologists (55.0–70.0%), neurosurgeons (66.7%), and orthopedic surgeons (68.8–81.3%). These differences were significant for a score of three (*p* = 0.017) and 10 (*p* = 0.041) and came close to statistical significance for a score of 15 (*p* = 0.064) (Table 3).

The correct interpretation of a “1b” ESCC score was more common among orthopedic surgeons (46.9%) than neurosurgeons (16.7%), medical oncologists (16.7%), and radiation oncologists (15.0%) (*p* = 0.043) and among physicians with ≤7 years (45.0%) or 8–14 years of experience (46.0%) than among those with >14 years (15.8%) (*p* = 0.016). The correct interpretation of a “2” ESCC score was also more common among neurosurgeons (66.7%) and orthopedic surgeons (50.0%) than among medical oncologists (22.2%) and radiation oncologists (20.0%) (*p* = 0.032).

### 3.4. Agreement with Statements

Agreement with the prognostic value of items included in the modified Tokuhashi score was ≥70%, except for the “number of bone metastases”, on which 48.8% of participants agreed (Table 4). Over 85% of the participants agreed with the prognostic value of the items included in the SINS score, and over 88% of the participants agreed with the recommendations from the NICE guideline, except for the one stating that all treating clinicians should participate in patient’s clinical assessment, on which 73.8% agreed (Table 4). 

Agreement with the usefulness of the number of spinal metastases to estimate life prognosis was higher among physicians treating ≥ 14 patients with MSCC a year (67.7%) than among those treating ≤ 7 (46.7%) or 8–13 (21.1%) (*p* = 0.006).

Among participants with ≥14 years of practice, agreement with the inclusion of pain assessment in the clinical evaluation of MSCC (79.0%) was lower than among those with ≤7 years (100%) or 8–13 (94.7%) (*p* = 0.037). No other differences in these variables were found across specialties, number of patients treated annually, years of experience, private/public sector, and hospital complexity.

## 4. Discussion

Participants in this study were specialists involved in the management of SCC, who volunteered for a study assessing their clinical practice. Bearing this in mind, results showing relevant deviations from the gold standard practice are striking. Only 40% of hospitals systematically included the ESCC classification in their radiological reports. Over 77% of specialists knew what the SINS was, but only 60% used it in routine practice, and a significant proportion of them misinterpreted the meaning of the SINS and ESCC scores. Since multidisciplinary collaboration is paramount for the successful treatment of MSCC, the fact that only 34% of hospitals had a Board for MSCC and only 37% had set up a multidisciplinary protocol for patients with MSCC is a grave cause for concern. These results are in line with those from other countries [12,13,14,19] and suggest that there is room for improvement in the management of SCC in routine practice.

Results from this study do not support the assumption that the public sector provides better care for patients with MSCC; in fact, data suggest the opposite in terms of the use of SINS and availability of specialized Boards. 

Some variations in results were also detected across hospital complexity and clinical experience. However, the most consistent differences were found among specialties. In general, physicians using interventional procedures (i.e., radiation oncologists, neurosurgeons, and especially orthopedic surgeons) were more familiar with the SINS and ESCC scores, used them more often, and were more accurate in interpreting their meaning than medical oncologists. This emphasizes the need for collaboration among specialists in routine practice.

Specialists with less than 14 years of experience interpreted the ESCC score more accurately than those with ≥14. This may suggest that more senior specialists rely less on a scale to assess the degree of spinal cord compression or that continued medical education should be reinforced for them, as is the case in other fields [20,21].

In general, there was a high degree of agreement with the prognostic value of most items included in the SINS and the Tokuhashi scores as well as with recommendations from the NICE guideline. However, many specialists did not use them in routine practice. This may reflect organizational obstacles in routine practice or disparity between knowledge and behavior.

Clinical experience was associated with agreement with some recommendations; specialists treating a higher number of patients with MSCC were more aware of the relevance of the number of spinal metastases to estimate life prognosis, whereas those with more years of practice tended to disregard the relevance of pain when assessing MSCC.

Due to sample size and the high number of comparisons, differences across participants’ characteristics should be interpreted with caution. This study aimed to identify potential gaps between current state-of-art recommendations and practice, in order to establish a hypothesis to be assessed in future studies with larger sizes and to assess whether actions should be undertaken to improve implementation of recommendations in routine practice. Therefore, at the design phase of this study, it was decided to prioritize sensitivity (i.e., identification of potential differences) and hence not to adjust results for multiples comparisons. 

This study has additional limitations. Participants were not selected randomly, but volunteered to participate in a study on SMD exploring their knowledge and clinical practice. The societies endorsing the study have over 16,000 members, but only 80 volunteered. It is likely that participants are those who are most familiar and concerned with MSCC. Additionally, this study gathered specialists’ reports on their own clinical practice, as opposed to data on their actual clinical practice. Therefore, results from this study may underestimate deviations from the gold standard practice in actual clinical practice. This is a cause for concern, since it might suggest that a sizable proportion of patients with MSCC may be receiving sub-optimal management in routine practice.

Future studies should confirm these results. A registry allowing surveillance, benchmarking, and analysis of variability of results, factoring in patients’ characteristics and treating physicians, was successfully implemented in routine practice for patients with back pain referred from the Spanish NHS to private facilities [22,23,24,25]. Bearing in mind the devastating consequences of MSCC, the suffering it causes, and the importance to ensure optimal treatment and coordination among specialists involved in treating this condition, similar strategies should be implemented to monitor the actual management of patients with MSCC in routine practice. Additionally, actions should be undertaken to further implement and expand the use of evidence-based recommendations for the diagnosis and treatment of patients with MSCC and the impact of such actions, both on the use of these recommendations in routine practice and on patients’ outcomes, should be assessed.

## 5. Conclusions

In conclusion, this study suggests that there is room for improvement in the routine management of patients with spinal metastatic disease. 

## Figures and Tables

**Table 1 cancers-15-02821-t001:** Characteristics of participants and their work settings.

Age ^1^		45.59 (10.7) [27–68]
Years of experience ^2^	As a specialist in training (*n* = 3)	4 (3; 5) [3–5]
	As a certified specialist (*n* = 77)	13 (7; 24) [1–34]
Specialty ^3^	Orthopedic Surgery	32 (40.0)
	Radiation Oncology	20 (25.0)
	Medical Oncology	18 (22.5)
	Neurosurgery	6 (7.5)
	Radiology	3 (3.8)
	Rehabilitatation	1 (1.3)
Number of patients with MSCC treated per year ^3^	≤7	20 (26.0)
	8–13	19 (24.7)
	≥14	38 (49.4)
Private/public sector ^3^	Only National Health Service	48 (60.0)
	National Health Service and private practice	25 (31.3)
	Only private practice	7 (8.8)
Hospital ownership ^3^	National Health Service	59 (73.8)
	Other govermental entities	14 (17.5)
	For-profit private entities	4 (5.0)
	Non profit private entities	3 (3.8)
Hospital management ^3^	Govermental (National Health Service or other governmental entities)	66 (82.5)
	Private	14 (17.5)
Complexity level of the hospital ^3^	Level 1	1 (1.3)
	Level 2	14 (17.5)
	Level 3	12 (15.0)
	Level 4	31 (38.8)
	Level 5	22 (27.5)
Hospital treating patients from the National Health Service ^3^	Yes	76 (95.0)
	No	4 (5.0)
Hospital has a protocol for clinical management of SMD ^3^	Yes	31 (38.8)
	No	41 (51.3)
	Unknown	8 (10.0)
Hospital has a multidisciplinary protocol for management of SMD ^3^	Yes	30 (37.5)
	No	18 (22.5)
	Unknown	32 (40.0)
Hospital has a Board for SMD ^3^	Yes	27 (33.8)
	No	53 (66.3)
Number of specialties represented in the Board ^3^	1	1 (1.3)
	3	7 (8.8)
	≥4	20 (25.0)

MSCC: Metastatic Spinal Cord Compression; ^1^: Mean (SD) [range]; ^2^: Mean (P25; P75) [range]: ^3^: *n* (%); see the text for details on differences found across private/public sectors.

**Table 2 cancers-15-02821-t002:** Description of clinical practice; *n* (%).

		All Participants (*n* = 80)	MO (*n* = 18)	RO (*n* = 20)	NS (*n* = 6)	OS (*n* = 32)	RX (*n* = 3)	RS (*n* = 1)
Is familiar with the Spine Instability Score (SINS)	Yes	62 (77.5)	9 (50.0)	17 (85.0)	4 (66.7)	28 (87.5)	3 (100)	1 (100)
	No	18 (22.5)	9 (50.0)	3 (15.0)	2 (33.3)	4 (12.5)	0 (0.0)	0 (0.0)
Uses SINS in routine practice	Yes, systematically	48 (60.0)	4 (22.2)	11 (55.0)	4 (66.7)	26 (81.3)	2 (66.7)	1 (100)
	Yes, occasionally	9 (11.3)	1 (5.6)	4 (20.0)	0 (0.0)	3 (9.4)	1 (33.3)	0 (0.0)
	No	23 (28.8)	13 (72.2)	5 (25.0)	2 (33.3)	3 (9.4)	0 (0.0)	0 (0.0)
Uses an outcome score in patients with MSCC	Yes, systematically	47 (58.8)	7 (38.9)	12 (60.0)	4 (66.7)	24 (75.0)	0 (0.0)	0 (0.0)
	Yes, occasionally	12 (15.0)	2 (11.1)	4 (20.0)	1 (9.4)	3 (9.4)	1 (33.3)	1 (100)
	No	21 (26.3)	9 (50.0)	4 (20.0)	1 (16.7)	5 (15.6)	2 (66.7)	0 (0.0)
Outcome score used in the hospital, if any	Varies across Departments	27 (33.8)						
	Tokuhashi	22 (27.5)						
	Tomita	10 (12.5)						
	Other	8 (10.0)						
	Do not know	13 (16.3)						
Imaging procedure used in the hospital to assess patients with MSCC	Full-spine MRI	57 (71.3)						
	MRI involved segment	18 (22.5)						
	CT segment involved	2 (2.5)						
	Other	3 (3.8)						
Radiological reports produced in the hospital include the ESCC score	Yes, systematically	14 (17.5)						
	Yes, occasionally	18 (22.5)						
	No	48 (60.0)						

MO: Medical Oncologists. RO: Radiation Oncologists. NS: Neurosurgeons. OS: Orthopedic Surgeons. RX: Radiologists. RS: Rehabilitation Specialist. See the text for details on the differences found across hospital complexity, specialties, and private/public sectors.

**Table 3 cancers-15-02821-t003:** Appropriateness of the interpretation of the scores in the SINS and the ESCC classification.

Correct Interpretation of the SINS Score (%)							
	All Participants (*n* = 80)	MO (*n* = 18)	RO (*n* = 20)	NS (*n* = 6)	OS (*n* = 32)	RX (*n* = 3)	RS (*n* = 1)
SINS score = 3	65.0	33.3	65.0	66.7	78.1	100	100
SINS score = 10	55.5	27.8	55.0	66.7	68.8	100	100
SINS score = 15	70.0	44.4	70.0	66.7	81.3	100	100
Correct interpretation of the ESCC score (%)							
ESCC score = 1b	30.00	16.7	15.0	16.7	46.9	67.7	0.0
ESCC score = 2	37.5	22.2	20.0	66.7	50.0	33.3	100

MO: Medical Oncologists. RO: Radiation Oncologists. NS: Neurosurgeons. OS: Orthopedic Surgeons. RX: Radiologists. RS: Rehabilitation Specialists. See the text for details on the differences found across specialties and years of experience.

**Table 4 cancers-15-02821-t004:** Agreement with the items to be assessed in patients with MSCCC (*n* = 80).

Item	N (%)
**Items related to prognosis**	
Oncologic prognosis	75 (93.8)
Number of spinal metastases	39 (48.8)
Score in tools (e.g., KPS or ECOGS)	73 (91.3)
Visceral metastases	67 (83.8)
Type/location of primary tumor	58 (72.5)
Degree of paresis	56 (70.0)
**Items related to spine stability**	
Mechanical pain	69 (86.3)
Type of bone lesion (lytic/blastic/mixed)	73 (91.3)
Spinal alignment	78 (97.5)
Degree of vertebral body collapse	76 (95.0)
Involvement of facet joints	77 (96.3)
**Items included in the recommendations from DG-75 NICE clinical guideline**	
Full spine MRI to assess compression	76 (95.0)
Neurologic examination	77 (96.3)
MRI to assess degree of compression	78 (97.5)
Clinical assessment of degree of compression by all treating clinicians	59 (73.8)
Assessment of limb strength and sensitivity	79 (98.8)
Assessment of pain	71 (88.8)
Assessment of sphincter control	79 (98.8)

KPS: Karnofsky Performance Score; ECOGS: Eastern Cooperative Oncology Group Score. See the text for details on differences found across physicians’ number of years in practice and physicians’ number of patients with SMD treated annually. See Appendix A for details on how the questions on the items included in the guideline were formulated.

## Data Availability

Data as well as the script for statistical analyses will be available under request (aroyuela@idiphim.org).

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
