# Peer review of "The Management of Metastatic Spinal Cord Compression in Routine Clinical Practice"

_cancers, 2023, doi:10.3390/cancers15102821_

Round 1

Reviewer 1 Report

on the downside of this paper ist the fact that it is difficult to extrapolate from one country to other systems. nevertheless i think this is a very realistic picture.

i think the message it sends is important and i favour to convey the message.

however i would like to see some changes, all of which related to the interpretation of these results. in particular:

i agree with the conclusions, that adherence to certain things are important nowadays when treating patients with secondary spinal malignancies. Yet there are clear differences between those things regarding their relevance. While it is "nice" to know and apply the SINS score, it is paramount to have all cases in a multidisciplinary tumorboard.

this is the key to being successful and providing the best of care to the patients.

i would like to see this message more prominent in the discussion and conclusions

Author Response

  • On the downside of this paper ist the fact that it is difficult to extrapolate from one country to other systems. nevertheless i think this is a very realistic picture. I think the message it sends is important and i favour to convey the message.

The authors thank the reviewer for these comments.

  • However i would like to see some changes, all of which related to the interpretation of these results. in particular: I agree with the conclusions, that adherence to certain things are important nowadays when treating patients with secondary spinal malignancies. Yet there are clear differences between those things regarding their relevance. While it is "nice" to know and apply the SINS score, it is paramount to have all cases in a multidisciplinary tumorboard. This is the key to being successful and providing the best of care to the patients. I would like to see this message more prominent in the discussion and conclusions

Following the reviewer’s comment, the updated version of the manuscript reads: 

·    Lines 47-48: Timely and efficient coordination among different specialists is paramount for appropriate treatment. [2] 

·    Lines 223-231: Participants in this study were specialists involved in the management of SCC, who volunteered for a study assessing their clinical practice. Bearing this in mind, results showing relevant deviations from gold standard practice, are striking. Only 40% of hospitals systematically included the ESCC classification in their radiological reports. Over 77% of specialists knew what the SINS was, but only 60% used it in routine practice, and a significant proportion of them misinterpreted the meaning of the SINS and ESCC scores. Since multidisciplinary collaboration is paramount for successful treatment of MSCC, the fact that only 34% of hospitals had a Board for MSCC, and only 37% had set up a multidisciplinary protocol for patients with MSCC, is a grave cause for concern.  

·    Lines 238-243: In general, physicians using interventional procedures (i.e., radiation oncologists, neurosurgeons and, especially, orthopedic surgeons) were more familiar with the SINS and ESCC scores, used them more often, and were more accurate in interpreting their meaning, than medical oncologists. This emphasizes the need for collaboration among specialists in routine practice.

·    Lines 277-281: Bearing in mind the devastating consequences of MSCC, the suffering it causes, and the importance to ensure optimal treatment and coordination among specialists involved in treating this condition, similar strategies should be implemented to monitor the actual management of patients with MSCC in routine practice.

Reviewer 2 Report

The authors describe in the paper „The Management of Metastatic Spinal Cord Compression in Routine Clinical Practice“ the knowledge and use of different scores in the assessment of SSC among different disciplines.

The main statments of the authors were: 1. Only about one third of the hospitals had a board and a multidisciplinary protocol, respectivel, for patients with MSCC. 2. Only 40% systematically included the ESCC classification in their radiological reports. 3. Over two thirds of specialists knew what the SINS was, but only 60% used it in routine practice. 4. A significant proportion of the specialists misinterpreted the meaning of the SINS and ESCC scores.

They conclude, that there is room for improvement in the management of SCC in routine practice.

In the opinion of the reviewer, a clinically very relevant topic is described. The manuscript clear and presented in a well-structured manner. The conclusion is consistent with the presented results. I recommend the acceptance in the present form.

Author Response

Point by point replies to your comments are detailed below (in italics):

The authors describe in the paper „The Management of Metastatic Spinal Cord Compression in Routine Clinical Practice“ the knowledge and use of different scores in the assessment of SSC among different disciplines.

The main statments of the authors were:

  1. Only about one third of the hospitals had a board and a multidisciplinary protocol, respectivel, for patients with MSCC.
  2. Only 40% systematically included the ESCC classification in their radiological reports.
  3. Over two thirds of specialists knew what the SINS was, but only 60% used it in routine practice. 4. A significant proportion of the specialists misinterpreted the meaning of the SINS and ESCC scores.

They conclude, that there is room for improvement in the management of SCC in routine practice.

In the opinion of the reviewer, a clinically very relevant topic is described. The manuscript clear and presented in a well-structured manner. The conclusion is consistent with the presented results. I recommend the acceptance in the present form.

The authors thank the reviewer for these comments.

Reviewer 3 Report

The ms by Pérez-Romasanta presents information about the practice on MSCC in Spanish hospitals.

The design of data gathering and performed analysis is done correctly, and of course, the conclusions are supported by the results.

My remarks are as follows:

Major

1. The research done is based on a questionnaire about routine practices in a disease that has many, as said, scores and scales to help in the decision-making process. That is no no-man's land where recommendations do not exist. It is just the confirmation of the lack of knowledge/lack of implementation of knowledge into routine practices. Therefore the scientific value of that research is scarce. If there would be no recommendations for MSCC, then the gathering of routine practices can be the base for future research and standard procedures. It is also not known whether that lack of following the standards results in worse patient care/outcomes.

2. What authors also state, 80 participants from 16000 is a little low number to perform a significant analysis.

3. The ms would be a good introductory part if it would be followed with a scheme/procedure for MSCC patients diagnosis and treatment to implement in the second step in Spanish / selected hospitals, and to analyze how that improved the practice.

Minor

4. First sentence: "The spine is the most common location for metastatic cancer [1,2]."

There is nothing that supports that in cited references:
[1] - Spinal metastases are the most common type of bone metastases with a prevalence of 30%–70% in cancer patients
[2] - The management of spinal metastatic tumors is a matter of increasing clinical importance, as 20-40% of cancer patients have evidence of vertebral metastatic disease at the time of their passing

5. The ms needs one thorough reading to correct grammatical/spelling/styllistic/punctuation mistakes.

Author Response

Point by point replies to your comments are detailed below (in italics)

The ms by Pérez-Romasanta presents information about the practice on MSCC in Spanish hospitals. The design of data gathering and performed analysis is done correctly, and of course, the conclusions are supported by the results. My remarks are as follows:

The authors thank the reviewer for these comments.

Major

  • The research done is based on a questionnaire about routine practices in a disease that has many, as said, scores and scales to help in the decision-making process. That is no no-man's land where recommendations do not exist. It is just the confirmation of the lack of knowledge/lack of implementation of knowledge into routine practices. Therefore the scientific value of that research is scarce. If there would be no recommendations for MSCC, then the gathering of routine practices can be the base for future research and standard procedures. It is also not known whether that lack of following the standards results in worse patient care/outcomes.

The authors fully agree with the reviewer in that the objective of this study was not to develop evidence or consensus-based recommendations, or to assess their impact on patient outcomes, but to assess whether the already existing evidence-based recommendations were actually followed in routine practice.

The authors feel that this information is valuable to decide whether actions should be undertaken to improve the implementation of evidence-based recommendations into clinical practice, in this field.

In accordance with the reviewer’s comment, the updated version of the manuscript reads:

  • Lines 47-55:

Timely and efficient coordination among different specialists is paramount for appropriate treatment. [2] To this end, several standardized methods have been developed [7-11]. However, whether these methods are actually used in routine clinical practice, is largely unknown. In fact, audits have reported inconsistencies between recommendations from Clinical Guidelines issued by the National Institute for Health and Care Excellence (NICE), and actual routine practice in the United Kingdom and Ireland [12-14].

The objective of this study was to explore the management of MSCC in routine practice in Spain, and to assess whether it followed available evidence-based recommendations.

  • Lines 258-262

This study aimed to identify potential gaps between current state-of-art recommendations and practice, in order to establish hypothesis to be assessed in future studies with larger sizes, and to assess whether actions should be undertaken to improve implementation of recommendations in routine practice.

  • What authors also state, 80 participants from 16000 is a little low number to perform a significant analysis.

The authors fully agree with the reviewer that the low rate of response of specialists, is a limitation of this study which, as the reviewer points out, is acknowledged and discussed in the text. 

In accordance with the reviewer’s comment, the updated version of the manuscript reads (lines 265-273):

This study has additional limitations. Participants were not selected randomly, but volunteered to participate in a study on SMD exploring their knowledge and clinical practice. The societies endorsing the study have over 16,000 members, but only 80 volunteered. It is likely that participants are those who are most familiar and concerned with MSCC. Additionally, this study gathered specialists’ reports on their own clinical practice, as opposed to data on their actual clinical practice. Therefore, results from this study may underestimate deviations from gold standard practice in actual clinical practice. This is a cause for concern, since it might suggest that a sizable proportion of patients with MSCC may be receiving sub-optimal management in routine practice.

  • The ms would be a good introductory part if it would be followed with a scheme/procedure for MSCC patients diagnosis and treatment to implement in the second step in Spanish / selected hospitals, and to analyze how that improved the practice.

The authors fully agree. 

In accordance with the reviewer’s comment, the updated version of the manuscript reads (lines 277-284):

Bearing in mind the devastating consequences of MSCC, the suffering it causes, and the importance to ensure optimal treatment and coordination among specialists involved in treating this condition, similar strategies should be implemented to monitor the actual management of patients with MSCC in routine practice. Additionally, actions should be undertaken to further implement and expand the use of evidence-based recommendations for the diagnosis and treatment of patients with MSCC, and the impact of such actions, both on the use of these recommendations in routine practice and on patients’ outcomes, should be assessed.

Minor

  • First sentence: "The spine is the most common location for metastatic cancer[1,2]." There is nothing that supports that in cited references:

[1] - Spinal metastases are the most common type of bone metastases with a prevalence of 30%–70% in cancer patients
[2] - The management of spinal metastatic tumors is a matter of increasing clinical importance, as 20-40% of cancer patients have evidence of vertebral metastatic disease at the time of their passing

The authors apologize for this error, and thank the reviewer for having detected it.

In accordance with the reviewer’s comment, the updated version of the manuscript reads (line 41):

Spinal metastases are the most common type of bone metastasis [1,2]…

  • The ms needs one thorough reading to correct grammatical/spelling/styllistic/punctuation mistakes.

The authors thank the reviewer for having drawn their attention on this. The text has been re-reviewed in depth by a native English speaker, and the errors detected have been resolved.